# Can Chemotherapy Negatively Affect the Specific Antibody Response toward Core Vaccines in Canine Cancer Patients?

**DOI:** 10.3390/vetsci10040303

**Published:** 2023-04-20

**Authors:** Paola Dall’Ara, Joel Filipe, Chiara Pilastro, Lauretta Turin, Stefania Lauzi, Elisa Maria Gariboldi, Damiano Stefanello

**Affiliations:** Department of Veterinary Medicine and Animal Sciences (DIVAS), University of Milan, Via Dell’Università 6, 26900 Lodi, Italy; paola.dallara@unimi.it (P.D.);

**Keywords:** dog, oncology, chemotherapy, core vaccination, antibody titration, canine parvovirus type 2 (CPV-2), canine distemper virus (CDV), canine adenovirus type 1 (CAdV-1), VacciCheck

## Abstract

**Simple Summary:**

The life expectancy of dogs has doubled in past decades. However, this positive effect has been accompanied by a concomitant increase in neoplasms. The aim of this study was to investigate the impact of antiblastic chemotherapy on the specific antibody response toward core vaccines in cancer-bearing dogs. Twenty-one patients with different types of malignancies were sampled before, during, and after different chemotherapy protocols to determine their actual levels of seroprotection against CPV-2, CDV, and CadV-1. No statistically significant changes in antibody titration emerged for any of the chemotherapy protocols used, suggesting that chemotherapy does not have an evident immunosuppressive effect on the post-vaccine antibody response.

**Abstract:**

The life expectancy of our pets has been getting longer in recent years due to new therapeutic opportunities, better nutrition, and better diagnostic approaches. This positive effect, however, has been accompanied by a concomitant increase in neoplasms, particularly in canine patients. Therefore, veterinarians inevitably face new issues related to these diseases, poorly or never investigated in the past, such as the possible side effects resulting from chemotherapy. The aim of this study was to investigate whether and how chemotherapy influences the antibody response against CPV-2, CDV, and CAdV-1 in dogs vaccinated before starting chemotherapy. Twenty-one canine patients with different types of malignancies were sampled before, during, and after different chemotherapy protocols to determine their actual levels of seroprotection against CPV-2, CDV, and CadV-1 by using the in-practice test VacciCheck. Differences related to sex, breed size, type of tumor, and chemotherapy protocol were evaluated. No statistically significant changes in antibody protection emerged for any of the chemotherapy protocol used, suggesting that, contrary to expectation, chemotherapy does not have a marked immunosuppressive effect on the post-vaccine antibody response. These results, although preliminary, may be useful in improving the clinical approach to the canine cancer patient, helping veterinarians fully manage their patients, and enabling owners to feel more confident about their pets’ quality of life.

## 1. Introduction

The life expectancy of pets has doubled in the past 40 years thanks to different life prolonging-factors, such as quality nurturing, regular veterinarian check-ups, and new therapeutic, diagnostic, and prophylactic approaches. This longer survival, however, has been accompanied by a higher frequency of diagnosis of various morbid conditions, among which neoplasms are the most common [1,2]. Veterinarians are therefore facing new issues related to these diseases, poorly investigated to date, such as the side effects of antineoplastic chemotherapy on the immune system [1,3,4]. The most common tumor types affecting canine patients are well known (e.g., mast cell tumors, lymphomas, hemangiosarcomas, transitional cell bladder carcinomas, soft tissue sarcomas), as are the chemotherapy protocols applied for their treatment (monochemotherapy or polychemotherapy) [5,6]. Notwithstanding, there are yet many aspects to be clarified in the field of immuno-oncology, particularly the impact of different tumor types and their therapies on the immune system [6,7,8]. In human medicine, it has long been known that cancer and chemotherapy increase susceptibility to opportunistic infections in patients, reactivate vaccine-preventable agents, and decrease the immune response after vaccination as a result of immune function impairment [7,9,10]. However, in canine oncology very little is known about these issues. One of the least understood aspects to date concerns the success of vaccination in canine cancer patients undergoing chemotherapy. Vaccines are the primary preventive tools for widespread and dangerous diseases both in human and veterinary practice. All international guidelines on good vaccination practices [11,12,13,14,15] classify pet vaccines as core and non-core. Those in the first category protect against some contagious, hazardous, and fatal diseases, and therefore are intended for all subjects, while those in the second category are suggested only for animals at risk of contracting specific diseases based on their lifestyle and geographical distribution, and therefore they are optional. The core vaccines for dogs are strongly recommended since they protect against three highly contagious, widespread, and often lethal diseases, i.e., canine parvovirus infection (caused by canine parvovirus type 2, CPV-2) [16,17,18,19], canine distemper (caused by canine distemper virus, CDV) [20,21,22], and infectious canine hepatitis (caused by canine mastadenovirus A, CAdV-1) [23,24].

In human medicine, chemotherapy and radiotherapy still represent fundamental curative treatments for patients affected by malignancies, and their increasing use in recent years has led to significant improvements in the survival of cancer patients. Nevertheless, these successful results may be associated with drawbacks that must be taken into account in the management of oncologic patients. One of the possible inconveniences of chemotherapy in human patients (and in children above all) is hematologic toxicity, which results in transient immunodeficiency involving mainly B lymphocytes. As already mentioned, antibody titers induced by vaccination could then be seriously compromised. However, the consequences for the immune system and vaccination appear to be highly variable as these are also influenced by factors other than chemotherapy [9,10].

The aim of this study was to investigate whether and how antiblastic chemotherapy impacts the antibody response against CPV-2, CDV, and CAdV-1 in cancer-bearing dogs vaccinated before starting chemotherapy.

## 2. Materials and Methods

### 2.1. Study Population and Study Protocol

The plasma samples used for this study were initially collected over a period of about one year (January 2021–March 2022) from 54 dogs with diagnoses of malignant neoplasm. The inclusion criteria are reported in Table 1. Eligibility for inclusion required mainly that the dogs were vaccinated with the core vaccines (CPV-2, CDV, and CAdV-1) and were undergoing antiblastic chemotherapy treatment in agreement with their cancer cytotype or histotype based on evidence from the literature given their cancer prognosis. Twenty-one dogs met the inclusion criteria and were therefore included in the study. All the other dogs were excluded from the analysis since they did not meet the inclusion criteria.

All patients included in the study underwent chemotherapy treatment in the dedicated room of the Veterinary Teaching Hospital (Department of Veterinary Medicine and Animal Sciences, University of Milan, Lodi, Italy). For each dog, key data were detailed: (1) sex and reproductive status: intact or neutered male or female; (2) age: adults, seniors, and geriatrics based on canine size according to the rule that small dogs live longer than large ones and vice versa [25,26]; (3) breed (purebred or crossbred) and animal size: small (<10 kg), medium (≥10–<25 kg), large (≥25–<45 kg), and giant (≥45 kg); (4) histologic/cytologic diagnoses of tumor; (5) type of dose-intensive chemotherapy (monochemotherapy or polychemotherapy and relative drugs) or metronomic chemotherapy; (6) number of antiblastic chemotherapy administrations and relative blood samples; (7) vaccination history, considering time elapsed since last vaccination: <1 year, ≥1 year–<2 years, or ≥2 years.

An extra group of 50 dogs in similar situations (vaccinated oncologic patients sampled before receiving any chemotherapy protocol) was used as a control group.

### 2.2. Sample Collection

All blood samples were collected in K_3_EDTA for the blood analyses required by the oncologic control, and residual aliquots were used for the antibody titrations specific for the core vaccines. According to the decision of the Ethical Committee of the University of Milan, residual aliquots of samples collected with the informed consent of the owners can be used for research purposes without any additional formal request for authorization (EC decision 29 October 2012, renewed with the protocol n. 02-2016). For each patient, one blood sample was collected before the beginning of the chemotherapy, and then again before each other chemotherapy administration. A final sample was collected at the end of the chemotherapy protocol. Whole blood samples were centrifuged for 15 min at 1500× *g*, and plasma was collected, recorded, and then stored at −20 °C until use.

Once the sampling was completed, three samples were tested for each dog: (1) the initial sample collected before the beginning of chemotherapy (T0); (2) a sample collected exactly in the middle of the patient’s chemotherapy protocol (T1); (3) the final sample collected at the time of the last clinical examination one month after the end of chemotherapy (T2). In this way, 63 samples were tested in order to detect eventual changes in the antibody titers specific for core vaccines due to chemotherapy treatment.

### 2.3. Detection of Specific Antibodies by VacciCheck

Each plasma sample was analyzed using the in-clinic test Canine VacciCheck (Biogal, Kibbutz Galed, Israel, supplied in Italy by Agrolabo, Scarmagno, Italy), following the manufacturer’s instructions. The kit is a dot-ELISA-based rapid semi-quantitative system approved to measure the specific antibody titers (IgGs) against CDV, CPV-2, and CAdV-1. VacciCheck has high specificity and sensitivity for each microorganism and is approved to be used for both research and diagnostic purposes [11,15,27,28,29,30,31,32,33]. In this rapid test, the antibody concentration is defined by the color intensity of the resulting spots compared with a scale from 1 to 6. The S0 value was standardized by the manufacturer as being equivalent to an antibody titer of <1:8 for CDV, <1:20 for CPV-2, and <1:4 for CAdV-1, while an S value of 3 (S3) was defined as being equivalent to 1:32 for CDV, 1:80 for CPV-2, and 1:16 for CAdV-1. A value equal or higher than S3 indicates that the individual is protected against each of these three diseases (Appendix A). 

### 2.4. Statistical Analysis

A statistical analysis was performed using the Graph Pad Prism 9 program (GraphPad Software, La Jolla, CA, USA). Antibody titer data were transformed with Log_2_. The normality of the data was tested (D’Agostino–Pearson normality test, Shapiro–Wilk test, and Kolmogorov–Smirnov test), and the following non-parametric tests were used: a Friedman test (with a Dunn’s multiple comparison test), a Mann–Whitney test, and a Kruskal–Wallis test. Values at *p* < 0.05 were considered statistically significant. Tendency was considered at *p*-value < 0.1.

## 3. Results

At the end of the study (March 2022), a total of 188 plasma samples from 21 canine patients with different malignancies, and who had been treated with different chemotherapy protocols, were collected. Of these, 14 (66.7%) were females (1 sexually intact and 13 neutered) and 7 (33.3%) were males (5 sexually intact and 2 neutered). Collectively, 6 dogs (28.6%) were intact while 15 (71.4%) were neutered. Their ages ranged from 2 to 13 years, with 4 being adult (19.0%), 11 senior (52.4%), and 6 geriatric (28.6%) patients. Regarding their breeds, 16 dogs were of pure breed (76.2%) while 5 were crossbred (23.8%); there were no breeds more represented than others (two Border Collies and only one Doberman Pinscher, one Newfoundland Dog, one Pitbull, one Cocker Spaniel, one Beagle, one American Staffordshire Terrier, one Bull Terrier, one Bernese Mountain Dog, one Greyhound, one Bobtail, one Golden Retriever, one Rottweiler, one Belgian Shepherd, and one French Bouledogue). Regarding their size, 2 were small (9.5%), 10 were medium (47.6%), 6 were large (28.6%), and 3 were giant in size (14.2%). Of the 21 oncologic patients, 10 (47.6%) suffered from a lymphoma, and of these 6 (28.6%) from a multicentric B cell lymphoma, 3 (14.3%) from a multicentric T cell lymphoma, and 1 (4.7%) from a cutaneous lymphoma; the other 11 dogs suffered from a variety of neoplasms: 3 (14.3%) from mast cell tumors, 2 (9.5%) from cutaneous soft tissues sarcoma, 2 (9.5%) from transitional carcinoma of the urinary bladder, and the others (1 per type, 4.8%) from undifferentiated testicular neoplasm, splenic hemangiosarcoma, pulmonary histiocytic sarcoma, and undifferentiated renal neoplasm. These oncologic patients were all treated with a chemotherapy protocol: 12 out of the 21 (57.1%) were treated with monochemotherapy, 5 (23.8%) using lomustine, 3 (14.3%) using vinblastine, 2 (9.5%) using doxorubicin, and 2 (9.5%) using carboplatin, while the remaining 9 (42.9%) were treated with polychemotherapy, 6 (28.6%) using a cyclophosphamide, doxorubicin, vincristine, and prednisone (CHOP)-based protocol, 2 (9.5%) using a combination of thalidomide/cyclophosphamide, and 1 (4.8%) using LOPP. The number of chemotherapy administrations and the corresponding blood samples ranged from 3 to 18 (1 dog each, 4.8%), with a variety of other possibilities: 8 for 4 dogs (38.1%), 6, 7, and 9 for 3 dogs (14.3%), 10 for 2 dogs (9.5%), and 4, 13, 14, and 17 for 1 dog each (4.8%). Finally, regarding the time elapsed since their last core vaccination, 14 (66.7%) had been vaccinated <1 year before sampling, 6 (28.6%) received their last vaccination ≥1–<2 years earlier, and only 1 (4.8%) had been vaccinated more than 2 years earlier. The main characteristics of the 21 canine patients of this study are reported in Table 2.

### 3.1. Antibody Titers and Kinetics of Protection

For each of the 21 dogs in this study, 3 samples were tested (at the beginning, in the middle, and 1 month after the end of the chemotherapy protocol) in order to analyze the kinetics of the impact of the chemotherapy on measurable antibody levels over time. The results of these analyses are reported in Table 3.

Almost all the dogs maintained excellent protection against the three diseases over the time of the chemotherapy treatment, i.e., 20 out of 21 (95.2%) for distemper, 17 out of 21 (81%) for infectious canine hepatitis, and 16 out of 21 (76.2%) for parvovirus infection. A statistical analysis did not show any significant difference among the different timepoints for any of the three diseases, and consequently the measurable antibody levels did not seem to be negatively affected by the chemotherapy protocols. Distemper antibody titers were found to be even higher at T2 than at T0 and T1, with a slight decrease at T1 and a subsequent increase at T2 (Figure 1a). Most of the samples had values for distemper fluctuating between 1:32 and 1:64, and they never decreased below the protective threshold.

Different statistical analyses were performed taking into consideration different biological variables (i.e., type of treatment, dog size, sex, and age), and the obtained significant results are presented below. The antibody levels of the 21 individuals of this study were also compared with the antibody levels of the control group (vaccinated oncologic patients sampled before any chemotherapy protocol, *n* = 50), and no statistical differences were found.

### 3.2. Evaluation of Influence of the Type of Chemotherapy

The peculiar increase in antibody titers for distemper virus at T2 was particularly evident in the subjects treated with vinca alkaloids (vinblastine and vincristine, alone or in combination with other drugs in the CHOP and LOPP protocols), and the difference between T1 and T2 was statistically significant (*p*-value = 0.0476), probably due to the higher antibody titers at T2 after an initial decrease at T1 (Figure 1b). For the other two viruses (CPV-2 and CAdV-1), no statistically significant differences were observed.

The 10 patients treated with vinca alkaloids were then compared with the other 11 dogs treated with different molecules, considering their values at T1, and statistically significant differences (*p*-value = 0.0237) were identified between them (Figure 1c).

Even if the antibody titers against distemper virus in the patients treated with vinca alkaloids were significantly lower at T1 than those of the patients treated with other drugs, both would remain in the same protective range (1:32–1:64). This statistically significant difference suggests that dogs treated with drugs other than vinca alkaloids have higher antibody levels against distemper than dogs treated with vinca alkaloids, at least halfway through their chemotherapy treatment, while at the end of chemotherapy (T2) the distemper antibody titers increase to the extent that they became highly protective.

### 3.3. Evaluation of Influence of the Type of Tumor

No statistically significant differences in antibody titers were observed in the groups of dogs affected by lymphoma (*n* = 10) and by tumors other than lymphoma (*n* = 11) for any of the diseases, even if a tendency toward a statistical significance was observed between T1 and T2 when parvovirus infection was considered (Figure 2).

An increase in antibody titers between T1 and T2 was noted in both groups (lymphoma and non-lymphoma), with a tendency toward significance (*p*-value = 0.0837) in the lymphoma group. For a more precise analysis, antibody titers were analyzed according to the type of immunophenotype of the multicentric lymphoma (B-cell or T-cell), but no statistically significant differences related to the type of lymphoma were found.

### 3.4. Evaluation of Influence of the Dog Size

Patients were also analyzed according to their size and divided into two groups: small/medium dogs (<25 kg, *n* = 12) and large/giant dogs (≥25 kg, *n* = 9).

A statistically significant difference in antibody titers was detected only for CAdV-1. Specifically, large/giant dogs had statistically higher antibody titers than small/medium dogs at T0 (*p*-value = 0.0321) and T2 (*p*-value = 0.0105), while no statistically significant differences were found at T1 (Figure 3). In any case, all the values were equal or higher than the threshold values, and all the dogs were found to be protected, regardless of their size.

## 4. Discussion

Studies investigating the possible negative effects of chemotherapy on specific immune responses in veterinary oncologic patients are rare, but all seem to confirm the results of this study, suggesting that chemotherapy does not have a strong impact on the specific immune response, or at least that the impact is not as severe as might have been expected [6,8].

In a study by Henry et al. (2001) [8], 21 dogs with different malignancies and 16 dogs with lymphoma, all treated with chemotherapy, were analyzed in order to highlight any changes in antibody response against parvovirus infection, distemper, and rabies infections. The lack of significant changes detected in the antibody titers following chemotherapy allowed the conclusion that an established immunity due to previous vaccinations is not significantly compromised by the chemotherapy applied to treat oncologic dogs. Some years later, Walter et al. (2006) [6] conducted a study to evaluate the effects of two different chemotherapy protocols (doxorubicin and a multi-drug chemotherapy) on humoral immunity to de novo vaccination in 12 dogs with lymphoma and 9 dogs with osteosarcoma. Antibody titers after vaccination were not significantly different between the control group and the dogs treated with chemotherapy, suggesting once again that chemotherapy may have less of an impact on the immune response than might be expected. Finally, in 2014, Elias et al. [7] evaluated whether the same immune system impairment described in human medicine could be observed in small animals undergoing chemotherapy. The study was conducted on eight dogs with lymphoma treated with a CHOP-based protocol and correctly vaccinated against CPV, and on another eight naturally CPV-infected, symptomatic dogs. Blood was collected from the lymphoma group prior to the CHOP chemotherapy and during the protocol (weeks 3, 6, and 9), and at the same timepoints samples were collected from the control group. The authors concluded that there was no evidence of a decreased immune response even after two chemotherapy cycles, indicating that the previously established immunity against CPV was not significantly impaired.

In this study, the dogs that underwent chemotherapy with vinca alkaloid showed an unexpected increase in antibody titers against distemper at the end of the therapeutic protocol (T2), following a transient decrease in T1. Although there is a lack of studies in the veterinary literature on the positive effects of chemotherapy with vinca alkaloids on the immune system, in human medicine some studies report such an effect. A study by Gameiro et al. (2011) [34] has reported that following the administration of cisplatin and vinorelbine (the latter is a vinca alkaloid) to mice with non-small-cell lung cancer (NSCLC), there was an immediate (around 1–2 days) immunosuppressive effect on the T lymphocytes, but this was followed (3–4 days later) by a strong increase in the lymphocyte population, and particularly in the Treg subpopulation. It is known that T helper lymphocytes act as regulators and can promote the maturation of plasma cells and consequently boost antibody synthesis. Further investigations are needed to determine whether the increase in the Treg subpopulation also occurs in dogs, and whether it accounts for the increase in distemper antibody titers observed at T2 in dogs treated with a similar vinca alkaloid. Moreover, this could indicate a recovery of the immune system once chemotherapy is completed, helped by the “stimulating” effect of the alkaloids used in the chemotherapy protocol. It is not clear why only distemper antibodies show such an increase, and not other antibodies directed toward other viruses.

The splitting of the oncologic dogs into two groups (lymphoma and tumors other than lymphoma) was based on the study by Elias et al. [7], in which the authors suggested that the variations in antibody titers observed in cancer patients mostly depended on the nature of the tumor itself rather than on other variables. The increase in antibody titers toward CPV noted at T2 in both the lymphoma and non-lymphoma groups could also be explained by the environmental persistence of the parvovirus, which exhibits strong resistance. It is possible that the patients had somehow come into contact with the pathogen in the environment, resulting in the stimulation of a specific antibody response. The tendency toward significance observed in the lymphoma group could be related to the fact that this group included most of the dogs that had reached the final stage of sampling, which was performed about a month after the end of the chemotherapy protocol. It is therefore very likely that by that time the immune system was recovering itself. A very interesting feature is that the patient who showed the highest T2 antibody increase was particularly young (2 years old), and thus of a very different age compared with the other patients in the group (all senior or geriatric), and this could have helped in the immune recovery. Even in absence of statistically significant differences related to the type of lymphoma (B- or T-cell), a B-cell neoplasm was generally linked to a major decrease in antibody titers (one or two dilution factors), while the T-cell lymphoma showed a milder decrease. The most plausible explanation for this finding is that chemotherapy is more aggressive toward B lymphocytes, which temporarily cease to perform their main function (antibody production) until a new stimulation by T helper lymphocytes comes.

The statistically significant difference found in dogs of different sizes, with larger dogs having higher antibody titers against CAdV-1 than smaller ones, needs further investigation. Generally, small dogs have a better response after vaccination or natural stimulation than larger ones, and for this reason they are often referred to as high-responders; by contrast, many large or giant breeds often include a high number of non-responder dogs [5,11,15,33,35]. Since this difference was already present before treatment, and therefore was not induced by chemotherapy, it is possible that the immune response was further enhanced by the contact with CAdV-2. This virus is spread from dog to dog through coughing and, like CAdV-1, survives in the environment for a long time. It is therefore possible that after coming into contact with this virus (attending dog parks, boarding facilities, or grooming salons), the dogs may have developed a good cross-immunity [24,32].

Killey et al. [33] observe that still little is known about the effects of different diseases on vaccine-induced protection. However the antibody titration results of the individuals in our study are very similar to those obtained using the same in-clinic test in a previous study with healthy dogs [32].

Due to the small sample size enrolled in this study, however, these preliminary results need to be confirmed with further investigations on a larger cohort of patients with different neoplasms (above all, of the lymphoid and myeloid lines), and treated with other chemotherapy protocols and/or with a more uniform sampling for histotype/citotype neoplasms and chemotherapy protocols. It would also be interesting to investigate other features of the immune system (i.e., cell-mediated and innate immunity) in order to better understand the real effects of chemotherapy on the immune system, since an individual may have a very low antibody titration against a specific pathogen while still maintaining good protection based on the cell-mediated immunity, or even the innate immunity [11,33]. Moreover, VacciCheck is a laboratory test with good sensitivity (optimal for CDV, 100%) and specificity (optimal for CPV-2, 100%), and it is officially recognized for its reliability, and is the test most recommended by different international authorities and researchers from among all the in-clinic tests specific for dogs available on the market [36].

Finally, it would be interesting and useful to investigate whether chemotherapy treatment produces any adverse effects on a vaccine intervention performed during antiblastic therapy. Nowadays, performing vaccine prophylaxis during antiblastic treatments is generally not recommended by oncologists, and the testing of a patient’s antibody titers specific for core vaccines is not currently a common practice, especially in the early stages of chemotherapy protocols, when the adverse effects an individual patient may suffer as a result of the antiblastic protocol are unknown (e.g., severe febrile neutropenia or gastrointestinal alterations). This study demonstrates that protection against known pathogens is not dramatically altered by chemotherapy as immune memory seems to work reasonably well. However, since every oncologic patient has his or her own clinical and therapeutical history and his or her own individual immune response in terms of efficiency and durability, an antibody titer evaluation prior to beginning chemotherapy could be a major advantage, helping veterinarians choose whether to vaccinate in the short term or to postpone vaccination boosters without taking any risk.

## 5. Conclusions

This study represents the first in Italy aimed at assessing the effects of chemotherapy on the immune response specific for core vaccines in oncologic canine patients. The results of this study confirm those of the other few analogue veterinary studies and allow us to state that antibody titers against canine parvovirus infections, canine distemper, and infectious canine hepatitis are not negatively influenced by chemotherapy, at least in the medium to short term, and with the protocols and for the neoplasms analyzed.

These preliminary results represent a valuable starting point for future investigations. They are very useful for veterinarians wishing to comprehensively manage their oncologic patients, and we suggest avoiding booster vaccinations for dogs undergoing chemotherapy. In addition, these results could allow owners to be reassured about their pet’s quality of life during treatment, ruling out the risk of incurring dangerous and sometimes fatal viral diseases.

## Figures and Tables

**Figure 1 vetsci-10-00303-f001:**
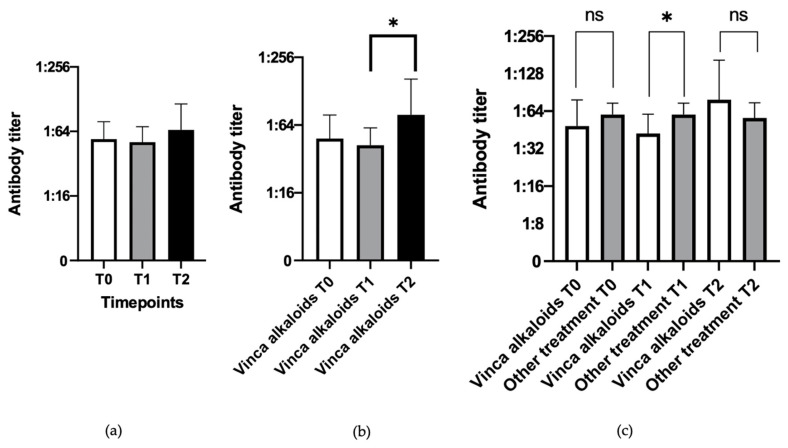
Antibody titers toward distemper at different timepoints (Kruskal–Wallis test). (**a**) all dogs treated with different chemotherapy protocols (*n* = 21 for each timepoint). (**b**) dogs treated with vinca alkaloids (*n* = 10 for each timepoint). (**c**) dogs treated with vinca alkaloids (*n* = 10) compared with dogs treated with other chemotherapy drugs (*n* = 11) at T1 (Mann–Whitney test). Statistically significant differences are indicated with *, and “ns” represents no statistical significant results.

**Figure 2 vetsci-10-00303-f002:**
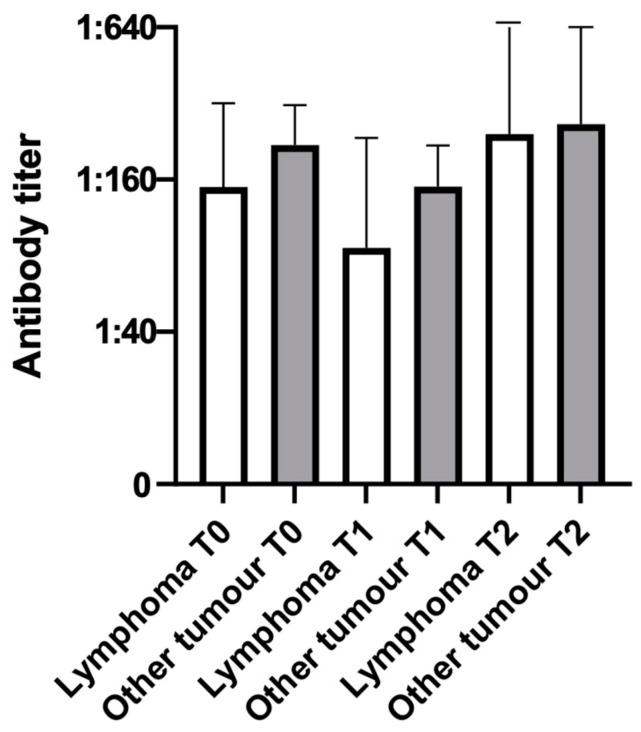
Antibody titers toward parvovirus infection at different timepoints in dogs with lymphoma (*n* = 10) and dogs with tumors other than lymphoma (*n* = 11).

**Figure 3 vetsci-10-00303-f003:**
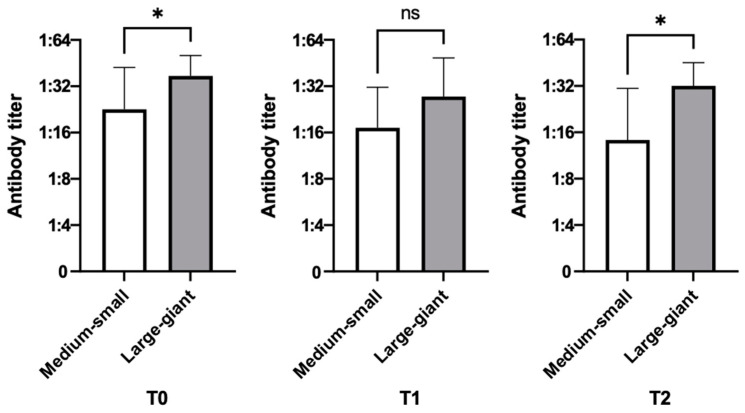
Antibody titers toward infectious hepatitis at different timepoints according to dog size (small/medium, *n* = 12; large/giant, *n* = 9). Statistically significant differences are indicated with *, and “ns” represents no statistical significant results.

**Table 1 vetsci-10-00303-t001:** Inclusion and exclusion criteria for the study.

Inclusion Criteria	Exclusion Criteria
Patient has a malignant neoplasm	Patient has a benign neoplasm or another concomitant pathology
Patient has to be treated with dose-intensive chemotherapy	Patient has a malignant neoplasm, but chemotherapy was not performed Chemoimmunotherapy
One blood sample was taken before beginning antiblastic chemotherapy	No blood sample was taken before antiblastic chemotherapy
Owner’s compliance	Owner’s noncompliance
Patient was vaccinated against CPV-2, CDV, and CAdV-1	Patient was not vaccinated against CPV-2, CDV, and CAdV-1
Patient undergoing at least two consecutive antiblastic chemotherapy treatments	Canine patient died before the end of antiblastic chemotherapy protocol

**Table 2 vetsci-10-00303-t002:** Characteristics of the 21 patients included in this study.

ID	Breed	Size	Sex	Age Category	Type of Tumor	Type of Chemotherapy	N° of Samples	Days from the Last Vaccination
1	Doberman Pinscher	Medium	NF	Adult	HS	L	7	263
2	Newfoundland Dog	Giant	IM	Senior	Lm	D	8	257
3	Crossbred	Medium	IF	Geriatric	TCUB	C	8	557
4	Pitbull	Medium	IM	Senior	UTN	C	4	160
5	Cocker Spaniel	Medium	NF	Senior	H	D	7	231
6	Border Collie	Medium	NF	Geriatric	Ly ^C^	L	8	86
7	Crossbred	Small	NF	Adult	Ly ^B^	CHOP	18	271
8	Beagle	Medium	NM	Senior	Ly ^B^	CHOP	10	270
9	American Staffordshire	Medium	NF	Senior	Ly ^B^	CHOP	3	860
10	Bull Terrier	Medium	NF	Adult	Ly ^B^	CHOP	10	120
11	Border Collie	Large	IM	Geriatric	BC	TC	7	663
12	Bernese Mountain Dog	Giant	IM	Senior	HS	L	13	257
13	Greyhound	Medium	NF	Senior	Ly ^T^	L	9	323
14	Bobtail	Large	NF	Geriatric	URN	TC	6	434
15	Crossbred	Large	NF	Senior	Ly ^T^	LOPP	14	109
16	Golden Retriever	Large	NF	Geriatric	Ly ^B^	CHOP	9	209
17	Rottweiler	Large	NF	Senior	Ly ^B^	CHOP	17	93
18	Crossbred	Large	NF	Geriatric	M	V	6	414
19	Belgian Shepherd	Large	NF	Adult	M	V	9	411
20	Crossbred	Small	IM	Senior	Ly ^T^	L	6	341
21	French Bouledogue	Medium	NM	Senior	M	V	8	423

IF = intact female; NF = neutered female; IM = intact male; NM = neutered male. HS = histiocytic sarcoma; Lm = leiomyosarcoma grade 3; TCUB = transitional carcinoma of the urinary bladder; UTN = undifferentiated tunica albuginea and vaginal process of the testis neoplasm; H = splenic hemangiosarcoma; Ly ^C^ = cutaneous lymphoma; Ly ^B^ = B cell lymphoma; HS = histiocytic sarcoma; Ly ^T^ = T cell lymphoma; URN = undifferentiated renal neoplasm; M = mast cell tumor. L = lomustine; CHOP = cyclophosphamide, doxorubicin, vincristine sulfate, and prednisone; TC = thalidomide/cyclophosphamide; LOPP = lomustine, vincristine sulfate, procarbazine, and prednisone; D = doxorubicine; C = carboplatin; V = vinblastine.

**Table 3 vetsci-10-00303-t003:** Antibody titers specific for core vaccines (CPV-2, CDV, and CaAdV-1) at T0 (before the treatment), T1 (during the treatment), and T2 (after the treatment) of the 21 dogs in the study. Protective results (equal or higher than the threshold values for each disease) are highlighted in pink (CPV-2), green (CDV), and light blue (CAdV-1).

	CPV-2			CDV			CAdV-1		
	Threshold Value: 1:80			Threshold Value: 1:32			Threshold Value: 1:16		
Dog N.	T0	T1	T2	T0	T1	T2	T0	T1	T2
**1**	160	160	80	64	64	32	32	32	4
**2**	320	160	160	32	64	32	32	16	32
**3**	160	160	160	64	64	64	32	32	32
**4**	320	160	160	64	64	64	16	8	8
**5**	160	160	160	64	64	64	32	32	32
**6**	160	40	80	64	32	64	32	8	16
**7**	160	40	160	64	64	128	32	16	32
**8**	40	40	20	32	32	32	4	8	4
**9**	40	40	40	64	32	64	16	16	16
**10**	160	20	160	64	32	64	16	16	16
**11**	320	160	160	64	64	64	32	32	32
**12**	320	320	320	64	64	64	32	16	32
**13**	320	320	160	64	64	64	32	32	16
**14**	160	80	160	64	64	64	32	16	32
**15**	160	160	160	64	64	128	64	64	32
**16**	160	160	160	32	32	32	32	16	16
**17**	320	320	320	64	64	128	64	64	64
**18**	160	80	80	64	32	128	32	32	32
**19**	320	160	160	64	64	256	32	32	32
**20**	320	160	160	64	64	64	32	32	32
**21**	160	160	160	16	32	32	32	16	16

## Data Availability

All the data are available within the article and from the corresponding author upon reasonable request.

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
