# Peer review of "Can Chemotherapy Negatively Affect the Specific Antibody Response toward Core Vaccines in Canine Cancer Patients?"

_vetsci, 2023, doi:10.3390/vetsci10040303_

Round 1

Reviewer 1 Report

Thank you for the opportunity to review this manuscript and for thoughtful consideration of this interesting research question. These data will be a useful contribution to the scientific literature in both oncology and immunology with real-world clinical applications for veterinary practitioners. Please consider the following points to strengthen the manuscript.

The aim of the study presented in the abstract (lines 22-23) differs from that presented in the manuscript (lines 68-69). The phrasing in the manuscript includes an assumption of negative impact, so the phrasing in the abstract is preferred by the reviewer. That said, the phrasing in the abstract indicates the response to vaccination would be assessed, when really the assessment here is whether chemotherapy had any impact on existing/measurable antibody levels. Please clarify and ensure that the two presentations of the study aim are consistent with one another.

Section 2.1 Please further describe inclusion criteria as it relates to vaccination history. Beyond timing of the most recent vaccination, was the timing and frequency of vaccination in these patients known or considered? If these are unknown, please be sure this point and the potential impact of different vaccination histories is addressed in the Discussion.

Lines 119, 125, 126, 251. Please confirm that the test kit detects antibodies to CAdV-1, rather than CAdV-2 as a proxy for protection against CAdV-1.

Line 186. See previous comment regarding study aim. I’m not sure it’s accurate to say that an immune “response for core vaccinesis being measured here. Rather, the impact of chemotherapy on measurable antibody levels is being recorded.

Line 199. Similar to previous comments. These data do not confirm “protection provided by core vaccines” rather measurable antibody levels.

Lines 229, 232 and 336. As above. The data do not determine “more” or “less” protection, merely measurable antibody levels.

Lines 261-270. Suggest moving this information to the Introduction as this nicely sets up the justification for the study. Then, consider crafting a statement that summarizes the findings of the current study and serves as a lead-in to lines 271-275.

Line 298. As above – “response to distemper” is vague and not quite accurate, please rephrase.

Discussion (general). Please include a brief analysis of the limitations of the study. Presumably two key limitations being the small sample sizes for each type of neoplasia and each chemotherapy protocol as well as the limited vaccine history (see earlier comment) – lines 353-359 begin to address this and should be moved to the Discussion section. In addition, there should be brief discussion of the pros/cons/limitations of titer analysis and what they do/do not mean (ie, they only measure humoral immunity).

Lines 361-362. Please rephrase, it is not clear what the recommendation is here – avoid administering vaccines around the time of chemotherapy? Or give the vaccine if it is indicated since the chemotherapy is not likely to impact the antibody levels? Either way, I think the latter is an important point to include if faced with an oncologic patient who is known to be at high risk for one of these infectious diseases and has no vaccination history.

Author Response

Reviewer n. 1

Thank you for the opportunity to review this manuscript and for thoughtful consideration of this interesting research question. These data will be a useful contribution to the scientific literature in both oncology and immunology with real-world clinical applications for veterinary practitioners. Please consider the following points to strengthen the manuscript.

R1-C1. The aim of the study presented in the abstract (lines 22-23) differs from that presented in the manuscript (lines 68-69). The phrasing in the manuscript includes an assumption of negative impact, so the phrasing in the abstract is preferred by the reviewer. That said, the phrasing in the abstract indicates the response to vaccination would be assessed, when really the assessment here is whether chemotherapy had any impact on existing/measurable antibody levels. Please clarify and ensure that the two presentations of the study aim are consistent with one another.

R1-A1. We thank the reviewer for this comment, and based on it we decided to rephrase the aim presented in the introduction section in order to reflect the same idea that is presented in the abstract, and that was the real aim of this study “The aim of this study was to investigate whether and how antiblastic chemotherapy impacts the antibody response against CPV-2, CDV and CAdV-1 in cancer-bearing dogs vaccinated before starting chemotherapy.”. Lines 78-80.

R1-C2. Section 2.1 Please further describe inclusion criteria as it relates to vaccination history. Beyond timing of the most recent vaccination, was the timing and frequency of vaccination in these patients known or considered? If these are unknown, please be sure this point and the potential impact of different vaccination histories is addressed in the Discussion.

R1-A2. The vaccination history of these dogs was known, and all of them had a frequency of vaccination similar and regular in time.

R1-C3. Lines 119, 125, 126, 251. Please confirm that the test kit detects antibodies to CAdV-1, rather than CAdV-2 as a proxy for protection against CAdV-1.

R1-A3. We confirm that the used test kit (VacciCheck) detects antibodies against CAdV-1.

R1-C4. Line 186. See previous comment regarding study aim. I’m not sure it’s accurate to say that an immune “response for core vaccines” is being measured here. Rather, the impact of chemotherapy on measurable antibody levels is being recorded.

R1-A4. The sentence “the specific immune response for core vaccines” was replaced with “the impact of chemotherapy on measurable antibody levels” – Lines (198-199).

R1-C5 Line 199. Similar to previous comments. These data do not confirm “protection provided by core vaccines” rather measurable antibody levels.

R1-A5. The expression “protection provided by core vaccines” was replaced by “measurable antibody levels” - Line 213.

R1-C6. Lines 229, 232 and 336. As above. The data do not determine “more” or “less” protection, merely measurable antibody levels.

R1-A6. “are more protected from distemper” was replaced with “have higher antibody levels against distemper” (Lines 246-247), and “more protected“ was replaced with “having higher antibody titers” (line 344).

R1-C7. Lines 261-270. Suggest moving this information to the Introduction as this nicely sets up the justification for the study. Then, consider crafting a statement that summarizes the findings of the current study and serves as a lead-in to lines 271-275.

R1-A7. The text present in lines 261-270 was moved to the introduction section. (Lines 68-77).

R1-C8. Line 298. As above – “response to distemper” is vague and not quite accurate, please rephrase

R1-A8. “positive effect on the antibody response to distemper” has been replaced by “increase in antibody titers against distemper”. – Lines 305-306.

R1-C9. Discussion (general). Please include a brief analysis of the limitations of the study. Presumably two key limitations being the small sample sizes for each type of neoplasia and each chemotherapy protocol as well as the limited vaccine history (see earlier comment) – lines 353-359 begin to address this and should be moved to the Discussion section. In addition, there should be brief discussion of the pros/cons/limitations of titer analysis and what they do/do not mean (ie, they only measure humoral immunity).

R1-A9. A brief analysis of the limitations of the study was present in the conclusion section, but have been now moved into the discussion (lines 358-366), and new pros/con/limitations had been added in the discussion (Lines 367-371).

R1-C10. Lines 361-362. Please rephrase, it is not clear what the recommendation is here – avoid administering vaccines around the time of chemotherapy? Or give the vaccine if it is indicated since the chemotherapy is not likely to impact the antibody levels? Either way, I think the latter is an important point to include if faced with an oncologic patient who is known to be at high risk for one of these infectious diseases and has no vaccination history.

R1-A10. This comment is absolutely correct. In general, performing vaccine prophylaxis is not recommended during antiblastic treatments (by oncologists) and to date no one tests individual patient's antibody titers. Especially in the early stages of chemotherapy protocols, when the individual patient's response to the protocol is unknown (e.g., the development of severe febrile neutropenia or the presence of gastro-enteric symptoms). So we would like to clarify that knowledge of the antibody titer prior to chemotherapy should be done and in any case the decision to perform or not to perform a vaccine booster will have to be evaluated on a case-by-case basis both based on the individual dog's response to chemotherapy treatment, the tumor's response or not to chemotherapy ( e.g., does it make sense to vaccinate a dog with progressive oncologic disease and poor prognosis with less than 3 months life expectancy? ) and based on environmental history, epidemiologic risk, and compliance with international laws ( i.e. rabies vaccination for kennel stay or relocation abroad). This information was also added into the discussion section (Lines 372-382).

Reviewer 2 Report

Paola Dall’Ara and colleagues studied the effect of chemotherapy to the specific antibody response in canine cancer patients toward CPV-2, CDV and CadV-1 vaccines. This study will provide evidence on if chemotherapy will induce immunosuppress to patients. The conclusion of this study is no significant difference has been noticed. This study could be a sound study with so many clinical samples and detailed information. However, important controls are missing to draw this conclusion.

Major

1. A control group which include canine patients with cancer received vaccination but without chemotherapy is missing. In the manuscript, the effect of different types of chemotherapy to patients were investigated. In this term, only the difference between chemotherapy can be observed but not the effect of chemotherapy to immune response. In this case, we still cannot answer the question of the title.

2. A mock control group, which include the antibody titer of healthy canine only received vaccine without malignancies or chemotherapy is preferred. This control will provide a baseline of regular immune response to the core vaccines, which will help to better elucidate the effect of cancer and chemotherapy to the patients’ immune system.

3. In the manuscript, only the type of tumor and the size of the canine were analyzed. More analysis about biological significance could be included, such as the age, the breed and sex.

4. The reason why vinca-alkaloids was analyzed specifically needs to be explained in the manuscript.

Minor

1. Although the numbers of patients of different biological groups were provided, in all figures, it’s better to provide the number of patients of each column either in the figures or in the legends.

2. In figure 2, only T1 is compare between vinca-alkaloids and other treatment, more time points are needed.

3. I would suggest the author combine Figure 1 and Figure 2, which are focusing on the effect of the treatment. And combine Figure 3 and Figure 5 and other biological grouping in another figure, which will be focusing on biological status. Just a reminder, Figure 4 is missing.

Author Response

Reviewer n. 2

Paola Dall’Ara and colleagues studied the effect of chemotherapy to the specific antibody response in canine cancer patients toward CPV-2, CDV and CadV-1 vaccines. This study will provide evidence on if chemotherapy will induce immunosuppress to patients. The conclusion of this study is no significant difference has been noticed. This study could be a sound study with so many clinical samples and detailed information. However, important controls are missing to draw this conclusion.

R2-C1. A control group which include canine patients with cancer received vaccination but without chemotherapy is missing. In the manuscript, the effect of different types of chemotherapy to patients were investigated. In this term, only the difference between chemotherapy can be observed but not the effect of chemotherapy to immune response. In this case, we still cannot answer the question of the title.

R2-A1. Unfortunately, it is not possible to have a representative control group that follow our study group over time (canine patients with cancer that received vaccination but without chemotherapy): It would be unethical not to propose chemotherapy treatment in patients with neoplasms that require it and do a follow-up with samplings over time. In any case, our initial samples (T0) are samples from oncological animals without treatment (the first sampling was always made before any treatment). This has represented our control and allowed us to evaluate each patient over time, avoiding different variables (such as different chemotherapy protocols, tumors, age, health status and so on) that could have complicated the interpretation of results. Nevertheless, we added a new group of animals (n=50) in the same situation (canine vaccinated patients with cancer but without chemotherapy, from our database): analyses on this control group didn’t show any differences in antibody titration values comparing with the 21 patients of our study. This part has been added in the text (Lines 107-108; 220-223).

R2-C2. A mock control group, which include the antibody titer of healthy canine only received vaccine without malignancies or chemotherapy is preferred. This control will provide a baseline of regular immune response to the core vaccines, which will help to better elucidate the effect of cancer and chemotherapy to the patients’ immune system.

R2-A2. A mock control group was not used, but in bibliography is possible to find studies with healthy populations that have been tested with the same in-clinics test (and between these also our paper published two month ago that analyzed more than 1,000 Italian dogs with VacciCheck – n. 32 of the references of the new document), and in the Discussion section we added a sentence describing how our data compares with the data from those studies (Lines 354-357).

R2-C3. In the manuscript, only the type of tumor and the size of the canine were analyzed. More analysis about biological significance could be included, such as the age, the breed and sex.

R2.A3. Thank you very much for this question. Due to the limited sample size (21 dogs in total), it was decided not to investigate the variable breed: in fact, it would not have been possible to obtain optimal groups for statistical analysis given the high heterogeneity of the samples (there were no breeds more represented than others, but only 2 Border Collie and one subject of all other breeds). On the contrary, age and sex were actually taken in consideration and statistical analyses were performed for these variables, but no significant result was obtained. A phrase was added in the results section about this (lines 218-220).

R2.C4. The reason why vinca-alkaloids was analyzed specifically needs to be explained in the manuscript.

R2.A4.  The division of our samples based on treatment (patients treated with vinca alkaloids and patients treated with a different protocol) was made because we noticed an influence of treatment over time. By examining each individual, we were able to identify that those with an increase in antibody titer were actually the ones treated with vinca alkaloids (this interesting result was confirmed after the statistical analysis).

R2.C5. Although the numbers of patients of different biological groups were provided, in all figures, it’s better to provide the number of patients of each column either in the figures or in the legends.

R2.A5. We thank the reviewer for this comment, the number of patients represented in each column was added in the respective legends.

R2-C6. In figure 2, only T1 is compare between vinca-alkaloids and other treatment, more time points are needed.

R2-A6. The respective graphs for T0 and T2 were added in figure 2.

R2-C7. I would suggest the author combine Figure 1 and Figure 2, which are focusing on the effect of the treatment. And combine Figure 3 and Figure 5 and other biological grouping in another figure, which will be focusing on biological status. Just a reminder, Figure 4 is missing.

R2-A7. We thank the reviewer for this suggestion, figures 1 and 2 were combined in one single figure, now called Figure 1. Instead, for Figures 3 and 5 (now called Figures 2 and 3) we think that should be better kept them separated, since the information provided by those graphs represents different variables and diseases (type of tumor and parvovirus infection for Figure 2, dog size and infectious hepatitis for Figure 3).

Round 2

Reviewer 2 Report

The authors have addressed the questions properly, and important results have been better elucidated in the manuscript. I would suggest the manuscript to be published in this Journal.